# Molecular, Electrophysiological, and Ultrasonographic Differences in Selected Immune-Mediated Neuropathies with Therapeutic Implications

**DOI:** 10.3390/ijms24119180

**Published:** 2023-05-24

**Authors:** Edyta Dziadkowiak, Marta Nowakowska-Kotas, Wiktoria Rałowska-Gmoch, Sławomir Budrewicz, Magdalena Koszewicz

**Affiliations:** 1Department of Neurology, Wroclaw Medical University, Borowska 213, 50-556 Wroclaw, Poland; 2Department of Neurology, The St. Jadwiga’s Regional Specialist Neuropsychiatric Centre, Wodociągowa 4, 45-221 Opole, Poland

**Keywords:** chronic inflammatory demyelinating polyneuropathy, Guillain–Barre syndrome, multifocal motor neuropathy, biomarkers, electrodiagnosis, ultrasound

## Abstract

The spectrum of immune-mediated neuropathies is broad and the different subtypes are still being researched. With the numerous subtypes of immune-mediated neuropathies, establishing the appropriate diagnosis in normal clinical practice is challenging. The treatment of these disorders is also troublesome. The authors have undertaken a literature review of chronic inflammatory demyelinating polyradiculoneuropathy (CIDP), Guillain–Barre syndrome (GBS) and multifocal motor neuropathy (MMN). The molecular, electrophysiological and ultrasound features of these autoimmune polyneuropathies are analyzed, highlighting the differences in diagnosis and ultimately treatment. The immune dysfunction can lead to damage to the peripheral nervous system. In practice, it is suspected that these disorders are caused by autoimmunity to proteins located in the node of Ranvier or myelin components of peripheral nerves, although disease-associated autoantibodies have not been identified for all disorders. The electrophysiological presence of conduction blocks is another important factor characterizing separate subgroups of treatment-naive motor neuropathies, including multifocal CIDP (synonyms: multifocal demyelinating neuropathy with persistent conduction block), which differs from multifocal motor neuropathy with conduction block (MMN) in both responses to treatment modalities and electrophysiological features. Ultrasound is a reliable method for diagnosing immune-mediated neuropathies, particularly when alternative diagnostic examinations yield inconclusive results. In overall terms, the management of these disorders includes immunotherapy such as corticosteroids, intravenous immunoglobulin or plasma exchange. Improvements in clinical criteria and the development of more disease-specific immunotherapies should expand the therapeutic possibilities for these debilitating diseases.

## 1. Introduction

Chronic inflammatory demyelinating polyneuropathy (CIDP) is the most common autoimmune neuropathy that can be monophasic, progressive, or relapsing. The etiology is unclear, with damage predominantly to the myelin sheath leading to demyelination and, with a prolonged process, axonal loss. A cellular and humoral response is involved in the autoimmune response [1,2]. The differential diagnosis of typical CIDP and variants of CIDP is widespread. The aim of many studies and clinical trials is to demonstrate the most common errors in the diagnosis of CIDP, as early diagnosis allows appropriate treatment to be implemented and disability to be avoided [3,4,5]. Many researchers in different regions of the world have highlighted the diagnostic errors in establishing a diagnosis of CIDP, hence the diverging information on prevalence and incidence. The prevalence is 1.0–8.9 persons per 100,000 thousand inhabitants per year and the incidence is 1.6 per 100,000 thousand persons per year [3].

There is a typical form of CIDP characterized by gradually increasing symmetrical muscle paresis of the proximal and distal limbs over a period of at least eight weeks with hypo or areflexia with sensory disturbances, weakness, less commonly cranial nerve involvement, optic disc edema and autonomic dysfunction.

A greater challenge is the diagnosis of the CIDP variants:distal CIDP (synonyms: distal acquired demyelinating symmetric neuropathy), 7–15% presents with sensory loss in the distal limbs as well as gait instability,multifocal CIDP (synonyms: multifocal acquired demyelinating sensory and motor neuropathy [MADSAM]; multifocal demyelinating neuropathy with persistent conduction block, Lewis–Sumner syndrome [LSS]; multifocal inflammatory demyelinating neuropathy), 4–14% involvement of asymmetric sensory and motor fibers, more often in the upper limbs,focal CIDP 4–14% involvement of one limb or nerve plexus (usually the brachial or lumbosacral plexus),sensory CIDP 3.5–14%—only sensory fibers are involved, characterized by gait ataxia, impairment of vibration and position sense and changes in cutaneous sensation,motor CIDP 4–9%—only motor fibers both proximally and distally [3,6,7,8,9].

The EFNS/PNS criteria (European Academy of Neurology/Peripheral Nerve Society guideline on diagnosis and treatment of CIDP) are the most commonly used in routine clinical practice and research and include clinical, electroneurography and adjunctive guidelines, which increase diagnostic sensitivity [7]. Doneddu and co-authors [10] compared the sensitivity and specificity of the 2021 European Academy of Neurology/Peripheral Nerve Society (EAN/PNS) diagnostic criteria for chronic inflammatory demyelinating polyradiculoneuropathy (CIDP) [7] with those of the 2010 European Federation of Neurological Societies/Peripheral Nerve Society (EFNS/PNS) [11]. The study demonstrated that the EAN/PNS criteria are more specific but less sensitive than the EFNS/PNS criteria. More extended nerve-conduction studies improved the diagnostic sensitivity of the EAN/PNS criteria maintaining a very high specificity.

In a clinical study carried out at the Department of Neurology Erasmus MC University Medical Center in Rotterdam, which enrolled approximately 122 patients from different centers in the country with a diagnosis of CIDP, it was shown that one-third of the patients were diagnosed with a different form of polyneuropathy and one-fifth of the patients had a completely different diagnosis [12]. A clinical trial in the USA found that approximately 50% of patients were misdiagnosed, 32% in Rotterdam and 68% in the UK [12,13,14]. Studies varied by study group, study duration and different healthcare systems. Studies have shown that diagnostic errors are similar in different parts of the world.

The diagnosis of the classic form of CIDP is generally unproblematic, the diagnostic challenge is the variants of CIDP. The distal form of CIDP requires differentiation from axonal neuropathies (e.g., metabolic-diabetic neuropathy) genetic or anti-MAG IGM neuropathy, POEMS syndrome (Polyneuropathy, Organomegaly, Endocrinopathy, Monoclonal plasma cell disorder, Skin changes). Anti-MAG antibody neuropathy is the most common IG M paraproteinemic neuropathy characterized by predominant sensory symptoms, ataxic gait, tremor upper limb with motor involvement and disability occurring late in the course of the disease. The disease is usually slowly progressive but may severely affect functional activities. The IgM paraprotein is typical for monoclonal gammapathy but also for MGUS, a lymphoproliferative disorder such as Waldenstrom’s Macroglobulinemia lymphoma or chronic lymphocytic leukemia CLL. The choice of treatment generally depends on the severity of the neuropathy. No adequate immunotherapy has so far been shown to be effective in anti-MAG neuropathy. Rituximab a chimeric monoclonal antibody against CD 20 +B lymphocytes has been assessed in two randomized controlled trials and nowadays is the most used in anti-MAG antibody neuropathy treatment. Despite being effective in less than half of patients with anti-MAG antibody polyneuropathy it seems to be the treatment of choice thanks to its safety and good tolerable profile. Additional potential therapies might include rituximab with associated chemotherapeutic agents such as chlorambucil or bendamustine. The BTK inhibitor ibrutinib in patients with anti-MAG neuropathy needs further confirmation in larger populations especially because of its good profile (oral administration and safety profile) [15,16,17]. POEMS is a paraneoplastic syndrome due to plasma cell neoplasm. The major criteria for the POEMS syndrome are polyradiculopathy, clonal plasma cell disorder (PCD), sclerotic bone lesions, elevated vascular endothelial growth factor and the presence of Castelman disease, others such as organomegaly, endocrinopathy, characteristic skin changes, papilledema, extravascular volume overload and thrombosis are minor features. Diagnoses are often delayed, as it can be mistaken for chronic inflammatory demyelinating polyradiculoneuropathy or a variant of POEMS syndrome, Castelman disease, which has no clonal PCD and no polyneuropathy. The diagnosis of the POEMS syndrome is made with three of the major criteria two of which must include polyradiculoneuropathy, clonal PCD and one of the minor criteria [18,19,20]. The multifocal form of CIDP MADSAM is differentiated from inflammatory polyneuropathies, genetic polyneuropathies such as HNPP and post-traumatic lesions. The motor form of CIDP is mainly associated with MMN and ALS, porphyria, inflammatory myopathies or neuromuscular junction disease. In the motor form of CIDP, an increase in clinical symptoms is possible with the administration of steroid therapy. The sensory form of CIDP is differentiated, among others, by metabolic diseases, paraneoplastic diseases and multisystem diseases [12,21].

Immune-mediated neuropathies are a broad category of diseases that differ in their time course, affected nerve fibers and disease associations. The authors undertook a literature review of the differential diagnosis of CIDP, with particular emphasis on multifocal motor neuropathy (MMN) and Guillain–Barre syndrome (GBS). The molecular, electrophysiological and ultrasonographic features of selected autoimmune polyneuropathies were analyzed, highlighting differences in diagnosis and ultimately treatment.

## 2. Methods

The authors searched the literature concentrated on CIDP, GBS and MMN with special reference to differential diagnosis. PubMed via MEDLINE and Google Scholar searches from early 1990 to 28 February 2023 were used. Reviews and published studies with subsequent verification of their reference lists for relevance to the subject were included. Conference abstracts and papers written in languages other than English were excluded. Keywords used: chronic inflammatory demyelinating polyneuropathy, CIDP, variants, Guillain–Barre syndrome, GBS, acute polyneuropathy variants, multifocal motor neuropathy, MMN, immunology, inflammatory process, nodal and paranodal antibodies, serum, cerebrospinal fluid, electrophysiological study, ultrasound study, and the treatment of immunological polyneuropathies. In addition to using individual keywords, the authors used PubMed Advanced Search Builder to find the most significant records. Three analysts (ED, MNK and WRG) worked separately to find the most relevant papers by sifting through the search engines.

The researchers worked individually and compiled a list of appropriate full-text manuscripts, followed by a discussion and comparison of the two lists. The 113 publications were the most relevant to the study and included in this review.

### 2.1. The Diagnostic Methods

#### 2.1.1. The Molecular Diagnostic

##### The Serum Biomarkers

In the previously published studies, antigens against myelin proteins P0, P2, PMP-22, tubulin and gangliosides GM1, LM1, and sulfated glucuronosyl paragloboside (SGPG) were identified as targets of immune attack in CIDP [22]. According to current guidelines, anti-MAG antibody testing is recommended for all patients with IgM paraprotein who fulfill the diagnostic criteria for CIDP (especially distal CIDP), as a high titer of anti-MAG antibodies (>7000 Bühlmann Titre Units, BTUs) would suggest a diagnosis other than CIDP [7]. Klehmet et al. [23] showed that long-term immunomodulatory treatment with intravenous immunoglobulin (IVIg) reduces the response of autoreactive T cells against PMP-22 and P2 antigens, which may be affected by the altered maintenance of CD8 and CD4 effector/memory T cell subsets toward a more anti-inflammatory immune state. The authors hypothesized that elevated PMP-22 and P2-specific T-cell responses may function as predictors of response to IVIg treatment [23]. Similarly, other studies have reported that patients responding to IVIg treatment show significantly greater T-cell responses against the myelin proteins PMP-22 and P2 compared to non-responders at baseline before IVIg treatment. In addition, responders demonstrated a decrease in the number of CD8+ effector memory T cells between baseline and follow-up to IVIg treatment, but there were no differences in CD4+ T cell subsets [24,25]. In 2020, Koike et al. [26] published a case report of a patient with CIDP who demonstrated circulating IgG antibodies against LM1, but not against NF-155, CNTN-1, GM1 and GD1b. Immunohistochemical analysis revealed the deposition of neoepitope C9, a component of MAC, on the compact myelin sheath. Macrophage infiltration with the presence of several CD68-positive cells in each fascicle was observed. In addition, complement deposition was noted in the internodes, which include most of the length of the myelin fibers. This case highlights the role of complement-dependent cytotoxicity in the pathogenesis of CIDP with anti-LM1 antibodies [26,27]. 

Currently, a new pathogenetic diagnostic category of autoimmune paranodopathies has been identified. It was the identification of antibodies directed against target structures of the paranodal region that allowed these neuropathies to be distinguished from other inflammatory neuropathies, including CIDP. The cell adhesion molecules contactin 1 (CNTN1) and contactin-related protein 1 (Caspr1) on the axonal side and neurofascin 155 (NF155) on the terminal myelin loops represent proteins important for the complex axoglial interactions configuring the nerve into three domains: nodes, paranodes and interstitials. The generation of antibodies against these axoglial regions conditions the disruption of the anatomy of the node of Ranvier modifying the neurophysiology of nerve conduction by affecting the saltatory conduction of myelin fibers without inflammation [28,29].

Prior infections are reported in up to 70% of patients with Guillain-Barre Syndrome (GBS) [30,31]. Therefore, molecular mimicry plays an important role in the pathomechanism of GBS, especially the axonal variant. Campylobacter jejuni lipooligosaccharide is similar to peripheral nerve membrane gangliosides. The passive immunization of rabbits with these ganglioside-like lipooligosaccharides led to similar clinical syndromes of flaccid tetraplegia, similar to the acute axonal variant of GBS motor neuropathy [32,33,34,35]. Anti-ganglioside antibodies have been shown to have various targets for peripheral nerves. Anti-GD1a antibodies bind to paranadol myelin, the nodes of Ranvier and the neuromuscular junction. GM1 and GQ1B antibodies bind to the peripheral nerve or neuromuscular junction. These different targets for the peripheral nerve are thought to condition the clinical heterogeneity of GBS presentation [36,37]. Specifically, certain gangliosides are more commonly related to specific GBS variants. Thus, anti-GM1 antibodies are found in the form of axonal motor neuropathy. Anti-GQ1B antibodies are associated with Miller–Fisher syndrome, while anti-GQ1b and anti-GT1a IgG antibodies can be demonstrated in the acute oropharyngeal variant of GBS. However, apart from the association of Miller–Fisher syndrome with anti-GQ1B antibodies, the sensitivity and specificity of all antibodies for specific subtypes is low to moderate yield for clinical utility [38,39,40]. Considering that some patients are seronegative for anti-ganglioside antibodies, more studies are required to explain the role of anti-ganglioside antibodies in GBS, either as a cause or an epiphenomenon.

In comparison to gangliosides, sulfatides such as 3-O-sulfogalactosylceramide represent a class of glycolipids with a sulfate group instead of neuraminic acid. In the peripheral nervous system, sulfatide is mainly located in the incomplete myelin of Schwann cells, but can also be found in the node and paranode. Whenever sulfatide is lacking or attacked by autoimmune reactions, the lateral loops and part of the nodes of Ranvier will be disorganized and, as a result, the myelin sheath may not function properly. Autoantibodies to sulfatide have been identified mainly in peripheral immune-mediated polyneuropathies (IMPN) with axonal damage [41,42]. Giannotta et al. [43] reported reactivity to sulfatide in only 1% of patients with CIDP.

Anti-GM1 antibodies in the IgM class can also be found in amyotrophic lateral sclerosis, GBS variants including acute motor axonal neuropathy (AMAN) and acute motor and sensory axonal neuropathy (AMSAN). Therefore, the presence of anti-GM1 antibodies is not specific or required for the diagnosis of MMN [44,45,46,47].

Serum IgM anti-ganglioside (anti-GM1) antibodies are prevalent in at least 40% of patients with multifocal motor neuropathy (MMN) [48] and, according to some reports, in more than 50–70% of patients [49,50]. Moreover, these patients commonly demonstrate a favorable response to intravenous immunoglobulin (IVIg) [51]. Anti-GM1 antibodies are suggested to play a pivotal role in the pathophysiology of MMN. It is a ganglioside predominantly expressed in the membrane of the motor axon and is involved in the clustering of ion channels in the nodal/paranodal region. Thus, the initial lesion is not targeted at the myelin sheath, but IgM anti-GM1 antibody binding causes the mislocalization and internalization of sodium and potassium channels, preventing the transmission of action potentials. The disruption of these ion channels, resulting in reduced propagation of the action potential, manifests itself in electrophysiological studies as a conduction block and reduced conduction velocity. In addition, a second disease mechanism is represented by the activation of complement, which mediates the formation of the membrane attack complex, compromising membrane integrity and causing axonal damage and loss. These complex mechanisms result in the detachment of myelin in the nodal and paranodal regions, the elongation of nodes and the disruption of ion channels determining altered membrane polarity and functional block of action potentials without actual demyelination [47,52,53,54,55].

Figure 1 shows the anatomy and molecular organization of the myelinated fiber and summarizes the characteristic biomarkers of immune-mediated neuropathies.

According to many authors, protein levels in the (CSF) are not an accurate marker. Although protein levels in the CSF are in the supporting criteria for CIDP, in studies carried out in selected groups of patients diagnosed with CIDP, protein levels were normal in the CSF and elevated protein levels were observed in patients with diagnoses other than CIDP. 

Most laboratories accept 0.45 g/L of protein in the PMR as normal, the implementation of values of 0.50 g/L for patients under 50 years of age and 0.60 g/L for those over 50 years of age has been proposed on the basis of clinical studies for patients with CIDP as an auxiliary criterion. Elevated protein levels in the CSF may result from a number of conditions, e.g., spinal stenosis, and diabetes mellitus. Cytosis in the CSF should not normally exceed 10 cells per mm^3^, but when it increases to approximately 50 cells per mm^3^, an infectious background such as HIV, Lyme disease, sarcoidosis or lymphoma should be excluded [3,12].

Acute and chronic inflammatory neuropathies are associated with an increase in neurofilament light chain (NfL) levels in cerebrospinal fluid (CSF) and plasma, but NfL is only approved as a prognostic biomarker in Guillain–Barré syndrome. T-tau in plasma is a new biomarker that could be a potential tool in the diagnostic assessment of patients with acute and chronic inflammatory polyneuropathies [59,60]. 

The protein concentration in the cerebrospinal fluid (CSF), however, does not distinguish the type of immune-mediated neuropathies. In MMN the cerebrospinal protein level is not so often elevated (9–18% cases) as in MADSAM or other atypical CIDP forms, although one of the supportive diagnostic criteria of MMN is the elevation of protein in the CSF (<1 g/L) [61,62,63].

## 3. The Electrodiagnosis

To minimize errors in electroneurophysiological examination, 5–8 motor nerves should be examined–ideally eight, as in the case of delicate sensory fibers, plexus or root involvement, lesions may not be registered on routine electroneurography. In order to increase the accuracy of the electroneurography examination, it is worth examining the contralateral side. CISP (Chronic immune sensory polyradiculoneuropathy) is an involvement of the dorsal sensory roots with preserved normal neurographic parameters of sensory and motor responses, but there are no data on whether the lesions are demyelinating or whether it is the sensory form of CIDP, which is why it is not classified as a variant of CIDP [7]. CISP is preserved in response to immunotherapy. A temperature too low of 30 degrees for the lower limbs and 33 degrees for the upper limbs can lead to prolonged latency and slower conduction velocity. A popular mistake is to interpret demyelination on electronegraphic examination when it is due to other pathologies. Even if a reduction in conduction velocity is found, it is not due to demyelination, but to axonopathy associated with a reduction in amplitude, which is most often due to the loss of fast-conducting fibers or the regeneration of immature nerve fibers. Symmetrical demyelination may be present in diabetic polyneuropathy, but if conduction blocks or increased temporal dispersion are present, these changes are not typical of diabetic polyneuropathy; according to EFNS criteria [7], conduction velocity slowing should not exceed 30% of the lower limit of normal. The electroneurographic result needs to be correlated with the clinical picture.

The finding of demyelination on electroneurography fulfilling the electroneurophysiological criteria, but in the absence of clinical signs for CIDP, it is advisable to look for other causes of demyelinating polyneuropathy. The interpretation of nerve conducting study results is particularly difficult when the response amplitude is reduced below 1 mV in which case the presence of conduction blocks cannot be excluded. Incorrect technical as well as potential marker positioning can cause deviations in electroneurograph examinations [3,6,12].

The electroneurographic result requires clinical correlation. Even unequivocal demyelination is not diagnostic for CIDP when clinical signs are absent, in which case a search for other causes of demyelinating polyneuropathy is indicated.

Electromyography and nerve conduction studies can be useful in distinguishing GBS from its mimics. Three to seven days after the onset of the first symptoms of GBS, early non-specific changes can be found on electrophysiological examination, i.e., absent or prolonged H-reflexes and/or F-wave latency, sural sparing pattern. In acute motor axonal neuropathy (AMAN), the examination usually shows a pattern of low, complex amplitudes of muscle action potentials or even inexcitable motor nerves, and a partial motor conduction block or complete conduction block can be observed in the nerve conduction study (NCS) of AMAN. This phenomenon is interpreted as “reversible conduction failure”. Complement is deposited in the nodes of Ranvier and paranodal regions on peripheral nerves. Subsequently, the nerves may develop Wallerian degeneration causing significant and long-lasting axonal damage, or may reverse, recognizing conduction failure. This phenomenon explains the relatively rapid recovery of some severely impaired AMAN patients. Acute motor and sensory axonal neuropathy (AMSAN) would show low-amplitude motor and sensory potentials. Miller–Fisher syndrome is more commonly reported with reduced or absent sensory nerve action potentials [64,65,66]. 

Pivotal neurophysiological criteria in the diagnosis of multifocal motor neuropathy (MMN) are the diagnosis of conduction block (CB) of motor fibers exclusively, with normal sensory conduction through the same segment in mixed nerves. The two most commonly involved nerves are the median nerve and ulnar nerve, in their forearm segments, not in the typical areas of compression. Additionally, the block is focal and occurs suddenly, and, at least in the earliest stages of the disease, motor conduction distal to the site of the block may remain normal [48,49]. The diagnosis of CB also depends on the nerve examined, as defining CB as definite in the tibial nerve requires a larger decrease in CMAP than in the forearm segment of the median nerve. A considerable challenge in the diagnosis of CB in MMN is the phase cancellation caused by temporal dispersion resulting in a spurious CB, a common occurrence in demyelinating neuropathies [67]. Currently, there are no reliable and reproducible techniques to assess proximal CB sites. Needle stimulation of motor roots is difficult to attempt with a significant probability of false-positive recordings and is not well tolerated. Many of the criteria for CB in MMN also include a limitation on the maximum acceptable temporal dispersion. Similarly, it is difficult to reliably determine CB when the distal amplitude of the evoked CMAP is less than 1 mV. Electromyography (EMG) almost always discloses significant chronic denervation and renervation of muscles supplied by nerves from CB, demonstrating that axonal degeneration is an important feature of MMN even from the earliest onset of the disease. There are reports of MMN with typical clinical features but without identified CB. A possible explanation for this is a very proximal or distal location of CB where routine electrophysiology is unable to detect the block. Another possibility is that conduction studies have only been performed in clinically diseased limbs, whereas CB can also be found in nerves innervating muscles with normal strength. There do not appear to be any significant differences in clinical symptoms and response to treatment between those with and without focal block [68]. 

The figures summarize the characteristic nerve conduction study findings for CIDP [Figure 2A], GBS [Figure 2B] and MMN [Figure 2C] in the ulnar motor study, recording abductor digiti minimi, stimulating wrist (A1), below groove (A2), above groove (A3), axilla (A4), and Erb’s point (A5).

## 4. Ultrasound

Ultrasound is increasingly being used in the diagnostic evaluation of CIDP and other polyneuropathies of different etiology. Ultrasound can detect nerve enlargement and changes in the nerve structure, which are common in CIDP. In particular, the cross-sectional area (CSA) and cross-sectional area variability of the nerves can be measured to assess for nerve enlargement. Additionally, a change in echogenicity, fascicle size, nerve vascularity, and epineurium thickness can provide clues to diagnosis and neuropathy activity [69,70].

CSA reference values for peripheral nerves and brachial plexus have been reported in various studies in the literature [71,72]. It may be useful in the differentiation of inflammatory neuropathies such as CIDP, where ultrasound can demonstrate nerve enlargement and increased vascularity; with compressive neuropathies such as thoracic outlet syndrome (TOS), where nerve compression or entrapment can be visualized with ultrasound; or inherited neuropathies, where diffuse nerve enlargement can be observed [73,74].

The cross-sectional area variability is a more complex parameter, and the differentiation of normal from pathological heterogeneity of CSA remains an important limitation of this modality of ultrasonographic examination. Several measures have been introduced in the literature to quantify the pathological heterogeneity of CSA. The first measure is the intranerve CSA variability, which is defined as the ratio of the maximal CSA to the minimal CSA for each nerve. The second measure is the internerve CSA variability, which is defined as the ratio of the nerve with the maximal intranerve CSA variability to the nerve with the minimal intranerve CSA variability for each patient. Additionally, a third measure, called the side-to-side difference ratio of the intranerve CSA variability, has been introduced, which is defined as the ratio of the side with the maximal intranerve CSA variability to the side with the minimal intranerve CSA variability for each nerve [75,76,77]. The fourth measure is the intraplexus CSA variability, which is defined as the ratio of the maximal CSA of the brachial plexus to the minimal CSA of the brachial plexus. The fifth measure takes into account the sum of the CSA in distal and peripheral segments [78]. These measures may provide a useful tool for quantifying pathological ultrasound changes in peripheral nerves in immune-mediated polyneuropathies. One of the examples of such tools is Bochum Neuropathy Ultrasound Protocol [79] designed to distinguish CIDP from MMN or MADSAM.

Nerve enlargement was found in 69–100% of CIDP patients [80,81] leading to the recognition of CIDP as one of the most common causes of leprosy, just behind hereditary motor and sensory neuropathies (types 1 and 3).

Studies showed that the CSA of the brachial plexus and the median nerve were the most adequate measurements to distinguish between CIDP and axonal neuropathy, with the diagnosis of CIDP most likely if there was an enlargement in at least two sites in proximal median nerve segments and/or the brachial plexus [73,82,83]. 

Eftimov et al. [27] found that ultrasound had a sensitivity of 97.4% and specificity of 78.9% compared to a sensitivity of 69.4% and 85.7% for NCS in the diagnosis of CIDP. Herraets et al. [84] found a sensitivity and specificity of the short sonographic protocol of 87.4% and 67.3%, respectively. There is lacing evidence of ultrasound sensitivity and specificity in comparison to MRI or nerve biopsy.

The relationship between ultrasonography findings and functional disability remains uncertain, according to several studies [69,76]. Some research has reported a connection between disease duration and nerve enlargement severity [85], while others have suggested that distinct ultrasonography patterns might indicate different injury mechanisms [69]. Specifically, swollen, enlarged, and hypoechoic nerves may indicate demyelinating insults, while hyperechoic atrophic bundles of fascicles may indicate axonal damage [69]. According to Rattay et al. (2017) [86], hypoechoic nerve enlargement can be reversible, suggesting that edema and acute inflammation may be the cause of hypointensity, while progressive axonal damage, peri- and endoneural fibrosis, and epineural scar tissue may contribute to the hyperintensity of the nerve fascicles and perifascicular tissue. 

Studies have shown that ultrasound can accurately diagnose CIDP, especially in patients when other diagnostic tests are inconclusive. It is still speculative, whether echo signal and enlargement patterns result from distinct pathologies associated with certain autoantibody specificities, and until now studies answering those questions are lacking [87]. 

MMN studies have shown a greater side-to-side intranerve variability than in CIDP [77]. In AIDP, in comparison to the CIDP, a different pattern of lesion has been found, which can be standardized by Bochum Ultrasonography Score giving a sensitivity of 90% and specificity of 90.4% in distinguishing those two types of autoimmunological polyneuropathies [88].

Ultrasound is a valuable tool in the diagnostic and monitoring process of CIDP [87]. The method, however, has some limitations. From a technical standpoint, it is operator-dependent and the visibility of nerves can be limited as ultrasound may not visualize all nerves in the body, especially those in proximal areas. Furthermore, while ultrasound can distinguish between different patterns of nerve involvement, it may not reliably differentiate between various types of neuropathies, such as axonal versus demyelinating neuropathies. The final limitation worth mentioning is the lack of standardization in performing and interpreting ultrasound examinations of peripheral nerves. This can lead to variability in results and make it challenging to compare findings across studies. Although the introduction of standardized protocols [79,84] could be a remedy for this problem.

## 5. Treatment

The main goal of CIDP treatment is to slow the progress of the illness, improve movement and reduce the disability. The first line of immunotherapy is the steroids or Immunoglobulins. Daily oral corticosteroid doses commonly used are prednisone 60 mg equivalent to methylprednisolone 48 mg slowly reduced over 6–8 months depending on clinical response and possible side effects. There is no evidence that a higher dose is more effective. An alternative to oral corticosteroid treatment is intravenous corticosteroids Solu-Medrol 500–1000 mg per day for 4 days per month for 6 months. Corticosteroids are not recommended as a first-line treatment for motor CIDP as they may intensify symptoms. In motor CIDP immunoglobulin should be considered as initial treatment. There are side effects of corticosteroid treatment such as osteoporosis, gastric ulceration, diabetes, cataracts, and arterial hypertension. Corticosteroids should be regularly checked for if the current dose is still required; potential side effects may predominate benefits for treatment in low disability disease. Immunoglobulin treatment is strongly recommended as a dose of 2 g/kg is divided over 2–5 days every 2–6 weeks. Unfortunately, not all patients respond to the first course, so clinical experts recommend a second course a few weeks after the first course. Most patients require immunoglobulins maintenance treatment with usual doses of 1 g/kg every 3 weeks (0.4–1 g kg every 2–6 weeks). If the treatment is effective and the patient is stable the dose can be reduced by 25% per infusion or the treatment interval lengthened. This can be conducted every 6–12 months for the first 2–3 years of treatment. There is no evidence of a difference in effectiveness between different immunoglobulin preparation for treating CIDP. Plasma exchange is a second way of treatment for CIDP. In five exchanges over two weeks, peripheral veins should be used. The research studies did not show differences between the induction treatment for plasma and immunoglobulin, but plasma exchange may be less well tolerated and more difficult to administer. Adding an immunosuppressant or immunomodulatory drug may be considered but there is no evidence to recommend any particular drug. Azatioprine, mycophenolate mofenil or cyclosporin may be added to immunoglobulin or corticosteroids as maintenance treatment. Cyclophosphamide, ciclosporin or rituximab may be considered in patients who are resistant to immunoglobulin, corticosteroids or plasma exchange [15,89]. For neuropathic pain or dysaesthesia tricyclic antidepressants, pregabalin, gabapentin or serotonin-noradrenaline reuptake inhibitors (duloxetine or venlafaxine) are recommended as first-line treatment. Improvement after immunotherapy is considered supportive for the diagnosis of CIDP. It has been shown that 85% of misdiagnosed patients who were treated as if they had CIDP felt better after immunotherapy even though only 19% of them demonstrated objective improvement and in most cases, they had an immune-mediated disorder. The most common pitfall was the interpretation of symptoms such as fatigue and pain as a response to treatment. The best way to improve the objective assessment of treatment benefits is using the Medical Research Council scale for measuring muscle strength, INCAT (Inflammatory Neuropathy Cause and Treatment) or I-RODS (Inflammatory Raschbuilt Overall Disability Scale), PGIC (Patient Global Impression of Change), CAP-PRI (Chronic Aquired Polyneuropathy Patient Reported Index). It is very helpful to assess the benefits of treatment [7,90,91]. 

Pathophysiologically, the autoimmune attack is directed at myelin components in demyelinating GBS (AIDP) and at Ranvier’s node, paranodal and juxtaparanodal regions, in axonal forms of GBS, as well as in Miller Fisher Syndrome (MFS). Underlying the immune process in the axonal forms is molecular mimicry between microbial antigens and axonal components. These anti-ganglioside antibodies, directed mainly to GM1 and GD1a in the axonal form of GBS and to GQ1b in MFS, cause axonal damage in nodal regions and at nerve terminals, resulting in conduction block, which may be reversible, with subsequent well-being, or alternatively, there may be axonal degeneration and worse clinical condition. The underlying basis for this difference in treatment outcome is unknown [92,93]. In addition to antibody-mediated attack, complement activation contributes to the pathological process by disrupting sodium channel clusters in Ranvier’s nodes, and activation of dendritic cells by Campylobacter jejuni lipo-oligosaccharides induces B-cell proliferation through the production of interferon 1 and tumor necrosis factor. In terms of therapeutic implications, there are presently no practical variations in treatment between the different GBS subtypes, where advantages have only been found with plasma exchange (PE) and intravenous immunoglobulin administration [93,94,95,96] The meta-analysis of data collected from randomized control trials for the PE compared to placebo showed moderate-quality evidence of increased chances to improve in disability at four weeks follow-up, increased chances to regain full muscle strength and decreased likelihood of severe motor disability at one-year follow-up [97]. Although there are no sufficient comparisons between IVIg and placebo in adults, the Cochrane last meta-analysis on the topic [98] presents moderate quality evidence that starting IVIg within two weeks from onset accelerates recovery as effectively as PE in severe cases.

The combined treatment (PE followed by IVIg infusion) in one randomized trial showed a non-significant trend toward improvement in comparison to treatment with only PE [99]. The first randomized, double-blind trial investigating the added value of a second intravenous immunoglobulin course in patients with GBS with a poor answer to the first course of IVIg showed that adding the second course of IVIg does not have a clinically meaningful benefit for recovery [100]. According to metanalyses, corticosteroids have no considerable impact on GBS [101,102,103].

The identification of patients with MMN is important as the majority of them can be successfully treated with IVIg. Subcutaneous immunoglobulin (SCIg) may be an option for IVIg, but the evidence is very inconclusive. Further trials are required to identify patients with MMN in whom IVIg withdrawal is possible and to affirm the effectiveness of SCIg as an alternative supportive therapy [104].

### Improved Clinical Criteria and the Development of More Disease-Specific

Immunotherapies should expand therapeutic options for these devastating diseases. The FcRn blocker efgartigimod, a humanized IgG1-derived Fc fragment, which competitively inhibits the FcRn, is currently under investigation in CIDP [105]. Moreover, the anti-human FcRn monoclonal antibody rozanolixizumab is currently being assessed in phase 2 trials in CIDP [106]. Eculizumab, a humanized monoclonal antibody, has not proven effectivity in randomized phase 2 trials yet in GBS patients [107]. Other randomized controlled trials estimating the effectiveness of various interventions versus placebo did not prove improvement after interferon beta-1a treatment [108] and brain-derived neurotrophic factor (BNDF) [109]. A randomized trial comparing cerebrospinal fluid filtration and PE showed no improvement after the first mentioned method in disability in GBS patients after four weeks [110]. The Chinese herbal medicine tripterygium polyglycolide, compared in a randomized trial with corticosteroids, showed very low-certainty evidence toward improvement after 8 weeks of treatment (but not at the standard 4 weeks) [111]. Given the role of the complement pathway in the pathogenesis of MMN, eculisumab has also been considered as a potential therapeutic strategy. In 2011, an open-label trial [112] using eculisumab was conducted in 13 patients with MMN, 10 of whom were simultaneously treated with IVIg. Accordingly, the data are promising with regard to the safety of the drug; however, the benefit was only marginal, with no objective measurable improvement. In addition, most patients required continuous IVIg therapy during eculisumab treatment, suggesting that the benefit of IVIg may be independent of complement activation. Promising experimental data suggest the possibility of using ARGX-117, a humanized, Fc-enhanced human IgG1 inhibiting anti-C2 antibody. The binding of anti-GM1 IgM to motor neurons triggers complement activation that is C2-dependent and is inhibited by ARGX-117, an antibody targeting C2, which may, therefore, be a potential therapeutic target for MMN [112,113].

## 6. Conclusions

Although adherence to the EFNS/PNS guidelines for CIDP would substantially diminish the proportion of misdiagnoses, there remains a small number of patients who fulfill these criteria for CIDP but may have an alternative diagnosis (‘true mimics of CIDP’), such as POEMS syndrome, CANOMAD (chronic ataxic neuropathy, ophthalmoplegia, immunoglobulin M [IgM] paraprotein, cold agglutinins, and disialosyl antibodies), and neurolymphomatosis. Pain, systemic symptoms, suggestive electrophysiological findings and/or the presence of a monoclonal protein in the serum should raise the suspicion of a CIDP mimic. Initial response to steroids or IVIG, over-reliance on CSF and electrophysiological findings may be confounding. These patients may experience a delay in making a correct diagnosis and initiating appropriate treatment, leading to significant disability and morbidity. Several studies have concentrated on polyneuropathies that mimic CIDP; our target was to review the most relevant molecular, electrophysiological and ultrasound differences of immune-mediated neuropathies, including CIDP, GBS and MMN.

According to the study, diagnostic errors often involved underestimation of proximal muscle weakness, lack of knowledge of CIDP variants or subjective perception of improvement after immunotherapy.

## Figures and Tables

**Figure 1 ijms-24-09180-f001:**
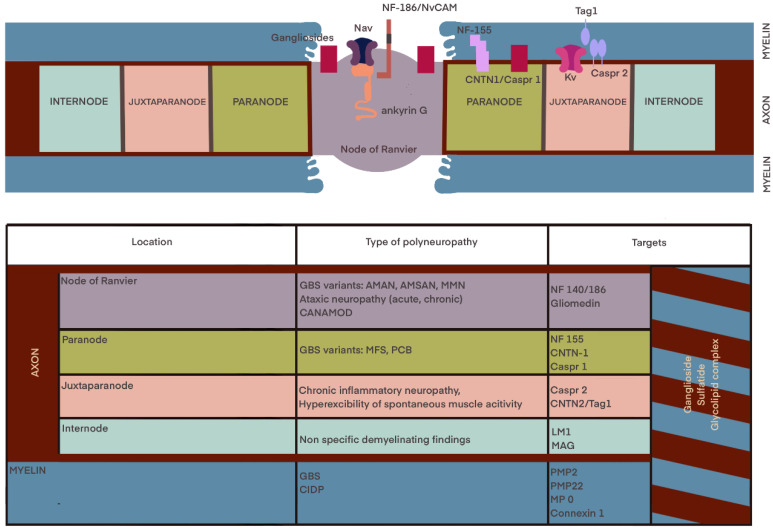
Anatomy and molecular organization of the myelinated fiber. Associations between the localization and type of polyneuropathy and specific site of neuronal damage according to [41,56,57,58].

**Figure 2 ijms-24-09180-f002:**
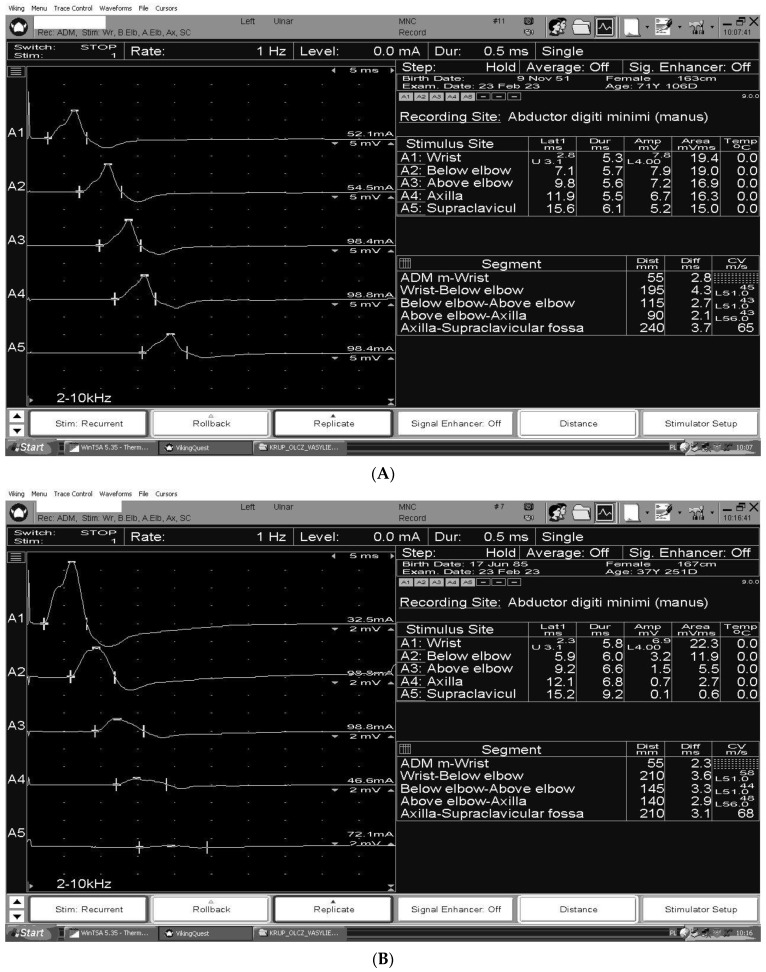
(**A**) Ulnar motor study in a patient with CIDP. Gradual, slight reduction in CMAPs amplitude between stimulation points at the wrist and Erb’s point. According to motor nerve conduction criteria the 2021 EFNS/PNS consensus guidelines for CIDP motor conduction block or slowing is not considered in the ulnar nerve across the elbow. (**B**) Multifocal conduction blocks in the ulnar nerve in the patient with GBS. (**C**) Temporal dispersion in the ulnar nerve at the Erb’s point stimulation in the patient with MMN.

## Data Availability

No new data were created or analyzed in this study. Data sharing is not applicable to this article.

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
