# Peer review of "Molecular, Electrophysiological, and Ultrasonographic Differences in Selected Immune-Mediated Neuropathies with Therapeutic Implications"

_ijms, 2023, doi:10.3390/ijms24119180_

Round 1

Reviewer 1 Report

Authors here describe an overview on some clinical, laboratory and neuropshysiological aspects of three form of inflammatory autoimmune neuropathies, such as CIDP, GBS and MMN.

I would have some concerns:

- one major issue is that authors, as to CIDP diagnosis, referred to 2010 EFNS/PNS criteria, but in my opinion it should be more correct to refer to more recent 2021 EAN/PNS criteria. So, all the section in the paper related to CIDP diagnosis (variants, neurophysiological features ad so on) should be updated taking in consideration these more recent criteria

- the second major issue is related to abstract: reading the title it seems that the focus of the paper is a review of some autoimmune neuropathies, but instead the Abstract seems focused on CIDP and, incidentally, GBS and MMN in terms of differences with CIDP. I think it shoul be re-formulated.

- some minor issues:

   -  more references in the therapy sections are needed (trial, real-world analysis, metanalysis)

  -  I think for a full completeness of the paper that authors should evaluate to add a section about anti-MAG polyneuropathy (or at least to highlight some critical issues in the differential diagnosis with CIDP).

Author Response

First of all, the authors would like to thank the Reviewer for all the comments. Every effort will be made to respond meticulously to the comments.
Recommended corrections have been made. Native has proofread the English language. 

The authors have updated the section of the article on the diagnosis of CIDP (variants, neurophysiological features, etc.), taking into account the 2021 EAN/PNS criteria.

The authors have reformulated the Abstract.  

References on future therapies have been completed. 

Information on anti-MAG polyneuropathy has been added and some critical points in the differential diagnosis with CIDP have been highlighted.

Reviewer 2 Report

Congratulations 

Author Response

The authors would like to thank the Reviewer for his acknowledgement. 

Reviewer 3 Report

Molecular, electrophysiological, and ultrasonographic differences in selected immune-mediated neuropathies with therapeutic implications.

This is an excellent review article compiled by the authors. In this article, the authors attempted to address the diagnostic challenge of the variants of CIDP and
other immune-mediated neuropathies with particular emphasis on Guillain-Barre syndrome (GBS) and multifocal motor neuropathy (MMN). They have analyzed the molecular, electrophysiological and ultrasonographic features of selected autoimmune polyneuropathies, highlighting differences in diagnosis and treatment. According to the study, diagnostic errors often involve underestimation of proximal muscle weakness, lack of knowledge of CIDP variants, or subjective perception of improvement after immunotherapy. I found this manuscript is well-written and valuable for the readers. However, the following are the suggestions to improve the manuscript further,

Specific comments,

1.    table 1 needs to be at the appropriate resolution and be easier to read. Please consider making it an appropriate resolution.

2.    What is the alternative diagnosis for CIPD?

3.    How can we overcome this problem in the near future?

Author Response

First of all, the authors would like to thank the Reviewer for all the comments. Every effort will be made to respond meticulously to the comments.

1. table 1 - resolution corrected

2. the differential diagnosis of CIPD has been completed

3. more treatment possibilities have been added. 

Round 2

Reviewer 1 Report

Authors correctly addressed the major issues highlighted in the previous revision.